# Poly(ε-caprolactone)-Based Graft Copolymers: Synthesis Methods and Applications in the Biomedical Field: A Review

**DOI:** 10.3390/molecules27217339

**Published:** 2022-10-28

**Authors:** Jean Coudane, Benjamin Nottelet, Julia Mouton, Xavier Garric, Hélène Van Den Berghe

**Affiliations:** 1Department of Polymers for Health and Biomaterials, Institute of Biomolecules Max Mousseron, UMR 5247, University of Montpellier, CNRS, ENSCM, 34000 Montpellier, France; 2Polymers Composites and Hybrids, IMT Mines d’Alès, 30100 Alès, France; 3EPF Graduate School of Engineering, 34000 Montpellier, France; 4Department of Pharmacy, Nîmes University Hospital, 30900 Nimes, France

**Keywords:** poly ε-caprolactone, graft copolymers, biodegradability, backbone functionalization

## Abstract

Synthetic biopolymers are attractive alternatives to biobased polymers, especially because they rarely induce an immune response in a living organism. Poly ε-caprolactone (PCL) is a well-known synthetic aliphatic polyester universally used for many applications, including biomedical and environmental ones. Unlike poly lactic acid (PLA), PCL has no chiral atoms, and it is impossible to play with the stereochemistry to modify its properties. To expand the range of applications for PCL, researchers have investigated the possibility of grafting polymer chains onto the PCL backbone. As the PCL backbone is not functionalized, it must be first functionalized in order to be able to graft reactive groups onto the PCL chain. These reactive groups will then allow the grafting of new reagents and especially new polymer chains. Grafting of polymer chains is mainly carried out by “grafting from” or “grafting onto” methods. In this review we describe the main structures of the graft copolymers produced, their different synthesis methods, and their main characteristics and applications, mainly in the biomedical field.

## 1. Introduction

Biocompatible and biodegradable synthetic polymers are gaining increasing attention for applications in the biomedical and pharmaceutical fields, such as tissue engineering, implantable medical devices, or drug delivery systems [1,2]. Most of these synthetic polymers are based on aliphatic polyesters, polyanhydrides, polyethers, polyamides, polyorthoesters, and polyurethanes. In particular, aliphatic polyesters are excellent candidates as they are hydrolyzed into their constituents, which are eliminated from the body through normal metabolic pathways. However, most properties of these polyesters (mechanical properties, ability to encapsulate drugs, rate of hydrolytic degradation, or biodegradation, etc.) depend on their chemical structure, which governs their hydrophobicity and crystallinity. 

PCL is one of these biocompatible and biodegradable synthetic aliphatic polyesters that is becoming increasingly important and that is FDA-approved for biomedical and pharmaceutical applications. Nowadays, poly lactic acid is undoubtedly the most used polyester for such applications because its properties can be easily modified by playing on its chirality. Although PCL is far from offering the versatility of poly lactic acid due to the lack of chirality in the PCL chain, more than 2000 papers referring to PCL are published each year (according to Web of Science). PCL has been particularly proposed for making sustained release drug delivery systems (DDS) due to its high drug permeability [3,4,5,6]. As a biomaterial, examples such as postoperative pleural and pericardial adhesions [7], sutures, dressings, and contraceptive devices [8] in the form of scaffolds, films, fibers, micelles, micro and nanoparticles [9] are mentioned in the literature. PCL is also known to undergo microbial and enzymatic degradation under external conditions [10,11,12,13,14], which is essential in the environmental field. The degradation of PCL in the human body is still a controversial topic of discussion, but as it is very hydrophobic and crystalline, it is degraded very slowly in vitro in the absence of enzymes and in vivo as well [15]. 

However, like most aliphatic polyesters (PLA, poly hydroxy butyrate, poly hydroxy valerate etc.), PCL is a non-functionalized polymer, except at the chain ends. It is therefore possible and easy to prepare diblock or triblock PCL-based copolymers, most of them incorporating PEG to decrease the hydrophobicity of PCL and obtain amphiphilic block copolymers. Many articles address the formation of these copolymers and their applications, especially in the biomedical field [16]. Specific structures such as star PCLs [17,18] or cyclic PCLs [19,20] are also described. The attachment of functional groups along an aliphatic polyester backbone, especially on PLA, is highly desirable to tailor properties such as crystallinity, hydrophilicity, mechanical properties, biodegradation rate, bioadhesion, and more generally biological activity [21]. In the same way, pending functional groups on the PCL backbone can be used to covalently attach polymer chains of biological interest to give PCL-based graft copolymers that are the focus of this review. 

Three chemical pathways are generally followed to yield such multifunctional PCL. The first one uses the polymerization of functional monomers, which is not the case for caprolactone and is therefore not relevant in this case. However, a few examples report on the direct formation of functional polyesters by polymerization of natural functional monomers. Some of them are based on malic acid [22,23,24,25,26,27,28], gluconic acid [29,30,31], or serine [32,33], giving functionalized aliphatic polyesters without prior chemical modification of the monomer structure. Nevertheless, in the case of PCL the absence of functional groups on the PCL backbone is a major drawback for obtaining graft copolymers with a PCL backbone. In a second case, the copolymerization of caprolactone (CL) with a functionalized co-monomer whose structure is different from that of CL is carried out. However, in this case, the backbone of the resulting copolymer is not strictly that of PCL and such examples will not be discussed in this review article. In the third case, starting from non-functionalized monomers, two general approaches are used to obtain functionalized aliphatic PCL: (i) the synthesis of a functionalized CL followed by its copolymerization with genuine CL and (ii) the direct chemical modification of a non-functionalized PCL (post-polymerization reaction). Both methods have been applied to obtain functionalized PCLs used as intermediate to finally obtain graft copolymers with a PCL backbone, sometimes referred to as “reverse” structure, in opposition with the “classic” structure (Figure 1) [34]. Such copolymers are reported and discussed in the frame of this review.

## 2. General Synthesis Methods of Functionalized PCL

### 2.1. Synthesis of a Functionalized Caprolactone

Copolymerization of a functionalized CL with a native CL is probably the most common method giving access to new functionalized PCL-based graft (co)polyesters. Functionalized CLs are the starting points for PCL-based graft copolymers, whose properties depend on the nature and proportion of functionalized CL in the copolymer. Many functionalized CLs are described in the literature: carboxyl-PCL [35,36], hydroxyl-PCL [35,37], amino-PCL [38,39,40,41,42,43], chloro-PCL [38,43], azido-PCL [38,43], propargyl-PCL [41,42], iodo-PCL [44], bromo-PCL and keto-PCL [45], 2-bromo-2-methylpropionyl-PCL [46], γ-(triethylsilyloxy)-PCL and γ-ethylene ketal-PCL [47], benzyloxy-PCL [32], 2-hydroxyethyl-PCL [48], and isocyanate-PCL [49]. Some of these functionalized copolymers are mentioned in a chapter of a book [50] and a review [51]. However, this versatile copolymerization approach has some drawbacks. Most of functionalized CLs are not commercially available; it is mandatory to synthesize and purify these lactones rigorously before polymerization, which is sometimes a challenging task. This is time consuming and involves multi-step chemical reactions with rather low yields. As a result, the overall yield of the synthesis is low, so the functionalized CL is quite expensive. On the other hand, several functional groups (e.g., epoxides, alcohols, or carboxylic acids) are not compatible with initiators such as aluminum and tin(IV) alkoxides frequently used for ring-opening (co)polymerization (ROP) of functionalized CLs. Although this drawback can be overcome by protecting the functional groups, the protecting groups must be removed after polymerization, which results in partial degradation of the polymer chain, especially when acidic conditions are required [50]. Moreover, the ROP of substituted CLs gives polymers with rather low polymerization degrees [40,41].

It is worth mentioning that some functionalized PCL can also be obtained by copolymerization of cyclic ketene acetals (CKA) with vinyl compounds, in particular vinyl ethers (VE) [52,53,54]. However, in these cases the backbone is not strictly PCL but a copolymer P(CKA-*co*-VE) of which only a part is PCL whose structure is not substituted.

In any case, in the literature very few of these functions are then used as anchor points for the grafting of polymer segments and therefore they do not give rise to graft copolymers.

### 2.2. PCL Backbone Modification

Along with the copolymerization method, post-modification of the PCL backbone is described. Basically, post-modifications of the PCL chain to yield a functionalized PCL do not involve the synthesis of a functional monomer. Therefore, the main advantages of post-modification over the functionalized monomers syntheses are ease, rapidity, and versatility. However, post-modification of polyesters chains, especially PCL, is quite difficult given the lack of reactive functions on their chains and the possible hydrolysis of ester groups during the post-modification reaction.

#### 2.2.1. Anionic Modification

Probably the most powerful method for functionalizing the PCL chain, described primarily in 2000, is the anionic chemical modification of PCL [55]. This method gives access to a multitude of substituted PCLs [39,40,49,56,57,58,59,60,61]. This direct post-polymerization method is based on a two-step one-pot anionic reaction in α-position of the carbonyl of PCL. A macropolycarbanion (PCL^−^) was obtained in the presence of lithium diisopropylamide (LDA), followed by electrophilic substitution, according to the reaction scheme in Figure 2.

The main characteristics of this method were described by Ponsart et al. [55]. Compared to the copolymerization of a functionalized caprolactone, the main advantages are the short reaction time, the functionalized PCL being obtained in one day, and the numerous possible substitutions. On the other hand, it appears that the main drawbacks of this method are (i) the breaking of some ester bonds in the PCL chain especially in the first step of the reaction, leading to a decrease in molar masses and (ii) the rather low substitution degree, typically under 20%, which has been attributed to the steric hindrance of the macropolycarbanion. However, the polymerization degree of the functionalized PCL obtained by this method is equivalent to that obtained by copolymerization of a substituted caprolactone with genuine caprolactone [40]. To increase the substitution degree, a higher ratio (LDA)/(monomer) is required, but the higher this ratio, the higher the proportion of chain breakings. Therefore, this ratio cannot be increased if a high molar mass is desired. The two parameters, substitution degree and molar mass, are correlated.

#### 2.2.2. Surface Modification

An advantage of the post-polymerization modification technique is that it can be applied to the surface modification of a manufactured object, without modifying the shape and the main properties (mechanical properties, degradation properties etc.) of the object. This is of particular interest when only surface properties need to be modified, especially in the biomedical field to improve the biocompatibility of particles or implantable devices and to promote endothelial cells adhesion and growth.

Some specific methods are well known to chemically modify the surface of a PCL-based device, including surface hydrolysis, aminolysis, and UV or plasma treatment. In general, polymer grafting is carried out in two steps: the first is the functionalization of the PCL surface and the second involves the polymerization of a monomer initiated by the new surface functions.

Hydrolysis in a basic medium is the most conventional method to functionalize a PCL surface. It results in formation of carboxylates and hydroxyl groups on the device surface [62]. The modification is studied as a function of hydrolysis time, temperature, and alkaline concentration with the aim of promoting endothelial cell (EC) adhesion and growth [63]. PCL films are subjected to variations in hydrolysis degrees using different pretreatment solutions to introduce various densities of carboxylate/hydroxyl groups onto the surfaces to modulate surface wettability and surface roughness. The hydroxyl groups on the hydrolyzed surface of a PCL film are reacted with 2-bromo isobutyrate bromide to prepare brominated PCL, which can initiate atom transfer radical polymerization (ATRP) of glycidyl methacrylate (GMA) to give PCL-*g*-PGMA [64]. The reactive epoxide groups of the grafted PGMA brushes were used for the direct coupling of cell-adhesive collagen and Arg-Gly-Asp-Ser (RGDS) peptides to improve the cell-adhesion properties of the PCL film.

Aminolysis is also used to functionalize the surface of PCL devices. Xiong et al. used 1,6 hexane diamine to obtain an aminated PCL, which then reacts with 2-bromoisobutyrate bromide to give an ATRP surface initiator for the polymerization of glycidyl methacrylate (GMA). Epoxy rings of the PCL-*g*-P(GMA) obtained can then react with gelatin to bind this compound to the PCL surface (Figure 3) [65]. A drastic increase in hydrophilicity is observed, as well as improved endothelial cells attachment and proliferation. An anti-thrombogenic profile was also observed.

However, strictly speaking, there is no chemical modification of the PCL chain upon hydrolysis or aminolysis, but chain breakings with the formation of carboxylates/hydroxyl/amine functions at the chain ends. Therefore, these copolymers are not real PCL-based graft copolymers.

Some authors have described the surface modification of PCL to attach growth factor or cell-adhesive biomolecules by UV photo-initiation [66]. A vapor phase is sometimes used for UV grafting [67,68]. UV-photoinduction was also used to graft poly gallic acid onto a previously hydrolyzed surface of PCL films to inhibit oxidative stress in epithelial cells [69]. Zhu et al. achieved grafting of poly methacrylic acid by the photo-oxidation of a PCL surface with hydrogen peroxide followed by UV irradiation at 30 °C of methacrylic acid [70]. The objective was also to immobilize gelatin on the surface to increase the surface cytocompatibility.

Plasma treatment is another method in the chemical modification of a polymer surface [71]. The surface modification depends on the gas (argon, oxygen, hydrogen peroxide, carbon dioxide, ammoniac), plasma power and exposure time [72]. Plasma treatment is widely described as a clean and environmentally friendly technique to improve the hydrophilicity, biocompatibility and biological performance of polymers for biomedical and tissue engineering applications [73,74]. Han et al. showed that on a PCL surface treated with argon plasma, human dermal fibroblast (nHDF) cell attachment density increased 60-fold after 1 min of treatment and more than 100-fold after 4 h of seeding compared to untreated PCL [75]. Oxygen plasma treatment allowed the grafting of polyethylene glycol mono acrylate onto a PCL membrane to prevent biofouling. The presence of grafted PEG on the surface was demonstrated by XPS technique and by a variation of the contact angle from 107° to 43°. PEG grafting led to a drastic reduction in fibroblasts adhesion. Oxygen plasma was also used to treat PCL electrospun nanofibers to allow initiation of graft copolymerization of acrylamide monomers [76]. The authors note an improvement in the antithrombogenicity of the grafted surface. The presence of hydrophilic functional groups was demonstrated by ATR-FTIR and contact angle measurements. Similarly, gamma irradiation of a PCL surface induced stepwise graft polymerization of acrylic acid (AA) and 2-aminoethyl methacrylate hydrochloride (AEMA) [77].

#### 2.2.3. Reactive Extrusion

Reactive groups such as maleic anhydride are grafted onto the PCL chain by reactive extrusion in the presence of dicumyl peroxide as initiator, leading to possible formation of PCL-based graft copolymers [78]. However, in general, reactive extrusion requires drastic experimental conditions, is often not or poorly controlled and is not applicable to many chemical reactions.

## 3. Case Study: Syntheses and Characterizations of PCL-Based Graft Copolymers

In this section we will focus on PCL-based graft copolymers issued from functionalized CLs or post-modification of PCL chain. The reactions are classified according to the nature of the synthesized copolymers.

### 3.1. PCL-g-PEG Copolymers

PCL-*g*-PEG is probably the most common of all PCL-based graft copolymers. The main applications stem from its amphiphilic structure, due to the hydrophilicity of the PEG segments and the hydrophobicity of the PCL chain. In all cases, a functionalized PCL chain is the starting point for the formation of PCL-*g*-PEG. Functionalized PCL can be made by either method, copolymerization of a functionalized CL with a genuine CL or chemical modification of the PCL chain, with some variations to these reactions. Without further indication, all these copolymers will be noted under the generic name PCL-*g*-PEG, regardless of the nature of the linkage between the PCL chain and the PEG.

According to the first method, a functionalized PCL is prepared by the copolymerization of α-chloro-ε-caprolactone (αCl-CL) with genuine CL. αCl-CL is probably the most widely used functionalized caprolactone for the preparation of functionalized PCL. Many examples of PCL-based copolymers discussed in this review are prepared using this functionalized caprolactone. Jérôme et al. describe some routes to obtain αCl-CL by oxidation of α-chlorocyclohexanone with *m*-chloro-peroxybenzoic acid. αCl-CL is then copolymerized with CL in toluene in the presence of 2,2 dibutyl-2-stanna-1,3-dioxepane (DSDOP) [79]. The resulting P(αCl-CL-*co*-CL) is then modified to P(α-azido-CL-*co*-CL) after substitution of the chlorine atom with sodium azide (Figure 4) [43]. Grafting of an alkyne-PEG onto poly (α-azido-CL-*co*-CL) is then carried out via a Huisgen CuAAC cycloaddition to give PCL-*g*-PEG. Some unreacted azide groups remain available for further grafting by another alkyne. This amphiphilic copolymer gives micelles whose CMC and diameter depend on the grafting degree of PEG (Table 1) [80].

The same poly (α-azido-CL-*co*-CL) also reacts with an α-acetal,ω-alkyne PEG to give a PCL-*g*-(α-acetal PEG) that is converted to PCL-*g*-(α-aldehyde PEG) by acid hydrolysis of the acetal group [81]. The presence of the aldehyde function allowed for further grafting of an amino sugar, typically a 2-aminoethyl mannose, onto the copolymer via reductive amination to provide a drug delivery targeting system. The bioavailability of mannose at the surface of PCL-*g*-(α-Mannose PEG) micelles was evaluated by surface plasmon resonance (SPR) (Figure 5) [81].

Another functionalized PCL, poly (α-propargyl-CL-*co*-CL) (PCL-yne), is also prepared by the ROP of a mixture of CL and α-propargyl CL in the presence of Sn(OTf)_2_. At the same time, αmethyl ether ω-azidoPEG1100 (MeOPEG-N_3_) was synthesized by mesylation of PEG monomethyl ether (MeOPEG) followed by nucleophilic substitution by sodium azide. CuAAC click reaction of PCL-yne on MeOPEG-N_3_ gave a PCL-*g*-PEG copolymer [42]. PCL-yne is also prepared via the anionic post-polymerization method by direct grafting of propargyl bromide on a PCL-based macropolycarbanion (PCL^−^) [82]. A thiolated MeOPEG (MeOPEG-SH) is then grafted to PCL-yne via photoradical thiol-yne chemistry in a manner that controls the hydrophilicity/hydrophobicity balance [83]. The main advantages of this thiol-yne “click” strategy are (i) grafting does not require metals, such as copper, which is of benefit for medical use and (ii) photo-chemical thiol-yne reactions are much faster than thermal thiol-yne or copper catalyzed reactions, thus preserving the integrity of the PCL chain. It was possible to tune the amphiphilicity of the copolymer to achieve better drug loading and cell uptake [84]. Using ω-substituted PEG-SH, various applications were envisaged: targeting properties with a cRGD-PEG-SH, imaging properties with a Fluorescein-PEG-SH or enzyme-responsiveness with a Meo-PEG-MMMcp-SH (MMPcp = Matrix Metallo Proteinase cleavable peptide) [56].

PCL-*g*-PEG is also the base of hydrogels and organogels, which are obtained by crosslinking PCL chains via a difunctional PEG. A α,ω-PEG dichloride is firstly prepared by reaction of PEG with triphosgene, and then reacted on PCL^−^ using the anionic method (Figure 6) [85]. The swelling degree of the cross-linked PCL-*g*-PEG depends on the proportion of PEG in the copolymer and the solvent (water, DMF, toluene) and ranges from 20% to 1230%. Lin et al. proposed the synthesis of a tridimensional PCL-*g*-PEG by copolymerization of caprolactone and PEG diglycidyl ether, but no evidence for the formation of a grafted structure instead of a linear one was demonstrated [86].

All these PCL-*g*-PEG have grafted segments in α position of the ester group. Another PCL-*g*-PEG structure was obtained using copolymerization of CL with a γ-substituted CL instead of a α-substituted CL. In this case a MeOPEG-γCL macromonomer was synthesized in a 5-step reaction (Figure 7) [87] and copolymerized with CL in dichloromethane in the presence of Al(OiPr)_3_ to give a poly [(MeOPEG-γCL)-*co*-CL] [88]. This method, which uses the copolymerization of a monomer substituted by a polymer chain with the same unsubstituted monomer, is often referred to as “grafting through”. This copolymer shows similar surfactant properties to PCL-*b*-PEG diblock copolymers and interfacial activity. It was also used for the preparation of stealth nanoparticles [89].

Another γ-substituted PCL-*g*-PEG is obtained by ring opening copolymerization of CL with 2-oxepane-1,5-dione or 1,4,8-trioxaspiro[4.6]-9-undecanone to give Poly (CL-*co*-(3-oxoCL)) [90] whose keto group then reacts with ω-amino MeOPEG to give P(εCL-*co*-(γMeOPEG-CL)), with the PEG grafted onto the PCL backbone via a stable ketoxime ether linkage (Figure 8) [91].

### 3.2. PCL-g-Poly (Meth)acrylic Graft Copolymers

#### 3.2.1. PCL-*g*-Poly Methyl Methacrylate (PCL-*g*-PMMA)

The grafting of methacrylic derivatives confers new and interesting properties, but the total biodegradability of the systems is lost because only the PCL chain is degraded. Many PCL-based graft copolymers containing poly(meth)acrylate segments are described in the literature. Poly (αCl-CL-*co*-CL) is used as macroinitiator for the CuCl/HMTETA (1,1,4,7,10,10-hexamethyltriethylenetetramine) mediated ATRP of methyl methacrylate (MMA), leading to the formation of a PCL-*g*-PMMA graft copolymer [79]. The molar mass of the PMMA segments, measured by SEC with PMMA standards, is about 29,000 g/mol.

PCL-*g*-PMMA copolymers have also been prepared from a PCL functionalized with pendant activated brominated groups. This brominated PCL is prepared by copolymerization of a bromo-substituted cyclic ester, γ(2-bromo-2-methyl propionyl)-ε-caprolactone (BMPCL) with caprolactone, by living ROP in the presence of Al(OiPr)_3_ (Figure 9) [46]. The pendent-activated alkyl bromide groups of the brominated PCL initiate the controlled ATRP of methyl methacrylate to give PCL-*g*-PMMA. In a second approach, BMPCL initiates the ATRP of MMA to give a PMMA macromonomer with a caprolactone chain end, which is then copolymerized with CL by ROP. The copolymers produced by the first synthetic route are found to have a more controlled molecular weight and narrow dispersity [46].

The same method using BMPCL was applied to the ROP of CL and BMPCL using MeOPEG as an initiator to give a MeOPEG-*b*-P(CL-*co*-BMPCL) copolymer. This copolymer is then used as macroinitiator for the ATRP of tertio-butyl methacrylate (*t*BuMA). After hydrolysis of tBuMA in methacrylic acid (MAA), a MeOPEG-*b*-(PCL-*g*-PMAA) copolymer was obtained (Figure 10) [92]. This amphiphilic copolymer is pH-sensitive, and can self-assemble in aqueous media into nanoparticles that incorporate and release ibuprofen in vitro with a higher rate as pH increases.

This MeOPEG-*b*-(PCL-*g*-PMMA) was then esterified by p-hydroxy benzaldehyde and reacted with cysteamine hydrochloride to yield MeOPEG-*b*-(PCL-*g*-P(MAA-Hy-Cys)) (PECMHC), which gives micelles in aqueous media that are cross-linked by oxidation of the cysteamine thiol groups. Loaded with doxorubicin these cross-linked micelles showed in vivo enhanced tumor accumulation because of their high stability in blood circulation and less DOX premature release (Figure 11) [93].

A PEG-*b*-(PCL-*g*-poly n-butyl acrylate) (PEG-*b*-(PCL-*g*-Pn-BuA)) was prepared using the same method. Molar masses of PBA segments were varied to modulate the stiffness of the copolymer [94]. The tumor penetration of self-assemblies has been shown in vivo to be dependent on their stiffness. The higher the molar masses of the PBA segments, i.e., the lower the stiffness, the higher the tumor penetration.

#### 3.2.2. PCL-*g*-P(2-N,N-Dimethylaminoethyl Methacrylate) (PCL-*g*-PDMAEMA)

PCL-*g*-P(2-(N,N-dimethylaminoethyl methacrylate) (PCL-*g*-PDMAEMA) was prepared by a combination of ROP and ATRP methods via the synthesis of BMPCL and its copolymerization with CL. The 2-bromo-2-methylpropionate group is used as a macroinitiator for the ATRP of dimethylaminoethyl methacrylate (DMAEMA) (Figure 12) [95]. PCL-*g*-PDMAEMA is an amphiphilic copolymer that forms nanoparticles with a very low CMC (8.1 × 10^−4^ gL^−1^ at pH = 7.4), lower than most amphiphilic diblock copolymers. It can simultaneously trap a hydrophobic drug and DNA. Furthermore, this cationic copolymer has shown excellent gene transfection efficiencies in both serum-free and serum-containing culture media. Similarly, by simultaneous copolymerization of DMAEMA and MeOPEGMMA (methoxyPEG methyl methacrylate), this team prepared a PCL-*g*-P(DMAEMA-*co*-MeOPEGMMA) [96,97,98]. In vitro cell viability studies show that PEGylation of PCL-*g*-PDMAEMA improves its biocompatibility. The excellent biocompatibility of these polycations by PEGylation is attributed to the ability of PEG to shield the positive charge of polycations. However, it reduces transfection efficiency at N/P ratios below 20 [97]. Lin et al. showed that these grafted copolymers contribute to a better SiRNA delivery efficiency in vitro and in vivo compared to block structures [99].

#### 3.2.3. PCL-*g*-P(N-Isopropylacrylamide) (PCL-*g*-PNIPAAm)

PCL-*g*-PNIPAAm was also obtained from P(αCl-CL-*co*-CL) that acts as macroinitiator of N-isopropylacrylamide (NIPAAm) ATRP, by electron transfer regenerated activators (ARGET ATRP), in the presence of CuCl_2_ and tris[2-(dimethylamino)-ethyl]amine (Me_6_TREN) [100] (Figure 13). PCL-*g*-PNIPAAm is an amphiphilic copolymer that can self-assemble into micelles whose size and CMC are dependent on the length of PNIPAAm segments. CMC increases from 6.4 to 23.4 mg/L as the mass fraction of PNIPAAm increases from 27.2% to 90.1%. Replacing P(αCl-CL-*co*-CL) with a brominated derivative, Massoumi et al. prepared PCL-*g*-PNIPAAm by ATRP of NIPAAm using P(αBr-CL-*co*-CL) as a macroinitiator [101]. The LCST of this thermoresponsive copolymer decreased from 32–33 °C for PNIPAAM to ≅30 °C for PCL-*g*-PNIPAAm with a grafting percentage of 18% in PNIPAAm.

Another method of access to MeOPEG-(PCL-*g*-PNIPAAm) is the synthesis of MeOPEG-P(CL-*co*-(*α*N3-CL)) by reaction of MeOPEG-P(αCl-CL-*co*-CL) on sodium azide, followed by a CuAAC click reaction on alkyne-terminal functionalized PNIPAAm [102] (Figure 14). Given the presence of the PNIPAAm segments, this copolymer is thermosensitive with a LCST around 17 mgL^−1^, lower than the one of PNIPAAm.

#### 3.2.4. PCL-*g*-P(MEO_2_MA-*co*-OEGMA)]-*b*-PEG-b-[PCL-*g*-P(MEO_2_MA-*co*-OEGMA) (tBG1) and PEG-*b*-[PCL-*g*-P(MEO_2_MA-*co*-OEGMA)]-b-PEG (tBG2)

To improve the hydrophilicity and impart temperature-response property to a PCL-*b*-PEG-*b*-PCL triblock, Wang et al. grafted poly (2-(2-methoxyethoxy) methacrylate)-*co*-oligo ethylene glycol methyl ether methacrylate) ((PMeO_2_MA-*co*-OEGMA) onto the PCL chains (Figure 15) [103]. The first step of the synthesis is a classical ROP of a mixture of CL and αCl-CL initiated by PEG with Sn(Oct)_2_ as catalyst to give the triblock P(αCl-CL-*co*-CL)-*b*-PEG-*b*-P(αCl-CL-*co*-CL). tBG1 was synthesized by the ATRP of a mixture of MEO_2_MA and OEGMA in the presence of CuCl, 2, 2-bipyridyl and P(αCl-CL-*co*-CL)-*b*-PEG-*b*-P(αCl-CL-*co*-CL) as macroinitiator. This copolymer shows a sol–gel transition around 35 °C, still reversible after more than 25 temperature fluctuations.

A variant of this synthesis was proposed by An et al., starting from the triblock MeOPEG-*b*-PCL-*b*-MeOPEG to improve its hydrophilicity and temperature sensitivity. A PEG-*b*-P(αCl-CL-*co*-CL)-*b*-PEG triblock was prepared by polymerization of a mixture of CL and αCl-CL in the presence of MeOPEG, followed by reaction with hexamethylene diisocyanate (HMDI) to form the triblock [104] (Figure 16). Segments of P(MEO_2_MA) and P(OEGMA) were grafted by the ATRP of a mixture of MEO_2_MA and OEGMA to give the final PEG-*b*-[PCL-*g*-P(MEO_2_MA-*co*-OEGMA)]-*b*-PEG (tBG2) copolymer. By adjusting the proportions of MEO_2_MA and OEGMA, the sol–gel transition of tBG can be adjusted around the physiological temperature 37 °C.

#### 3.2.5. Methoxypoly(ethylene glycol)-b-PCL-*g*-P(2-(guanidyl) ethyl methacrylate) (MeOPEG-b-PCL-*g*-PGEM) (PECG)

Another amphiphilic copolymer, (MeOPEG-*b*-PCL-*g*-PGEM) (PECG), is intended for vaccine preparation (Figure 17) [105]. It was synthesized from MeOPEG-*b*-P(CL-*co*-BrCL), with BrCL standing for γ-bromo-caprolactone. PECG was obtained by ATRP of (*t*-butoxycarbonyl) amino ethyl methacrylate in the presence of CuBr followed by deprotection of the Boc groups and guanidilation of the primary amines by S-ethylisothiourea. PECG self-assemblies in aqueous solution to form spherical nanoparticles. The authors demonstrate that dendritic cells activation and maturation, antigen cross-presentation, and immune responses in vivo depend on the presence of guanidyl moieties in the nanoparticles.

### 3.3. Other PCL-Based Graft Copolymers

#### 3.3.1. PCL-*g*-Polystyrene (PCL-*g*-PS)

While PCL/PEG and PCL/Poly(meth)acrylic graft copolymers are the most frequently described in the literature, other PCL-based graft copolymers can be found. PCL-*g*-PS copolymers are prepared by several methods. Riva et al. [43] described the formation of PCL-*g*-PS from P(N_3_-CL-*co*-CL). Propargyl bromoisobutyrate was grafted on N_3_-PCL-*co*-PCL via a cyloaddition reaction. The bromoisobutyryl groups were then used to initiate ATRP of styrene in anisole with CuCl/CuCl_2_-HMTETA as a catalyst according to a “grafting from” method. The PCL-*g*-PS copolymer was evidenced by its ^1^H NMR spectrum that showed new peaks at 7.2 ppm, which are typical of the aromatic protons of polystyrene.

PCL-*g*-PS was also obtained by CuAAC click reaction of an ω-N_3_-PS on propargylated-PCL using the “on to” method (Figure 18) [106]. Propargylated-PCL was prepared by copolymerization of propargylated CL with CL. The azido-polystyrene was prepared by CuBr-initiated ATRP of styrene, followed by NaN_3_ substitution on the brominated terminal group.

Finally, Nottelet et al. prepared PCL-*g*-PS from a iodinated-PCL (PCL-I) obtained by anionic substitution of the macropolycarbanion PCL^−^ with iodine [58]. PCL-*g*-PS was then obtained by iodine transfer polymerization (ITP) of styrene initiated by PCL-I according to the grafting “from” method (Figure 19) [107]. This procedure which uses anionic modification of PCL is far faster than the previous ones. Similarly, the authors prepared PCL-*g*-P(n-Butyl acrylate) and PCL-*g*-P(N,N-dimethyl acrylamide) by ITP of n-butyl acrylate (n-BuA) or N,N-dimethyl acrylamide (DMA). The molar masses of the branched PS (5000 g/mol), P(n-BuA) (2600 g/mol) and PDMA (3900 g/mol) were determined by SEC after alkaline hydrolysis of the PCL chain.

#### 3.3.2. PCL-*g*-PCL and PCL-*g*-Poly Lactic Acid (PCL-*g*-PLA)

Some methods described the preparation of polyester-*g*-polyester copolymers. In most cases, the objective is to obtain new properties while maintaining biocompatibility, biodegradability, and hydrophobicity. Therefore, these graft copolymers are most often composed of poly lactic acid and PCL.

PCL-*g*-PLA copolymers were recently prepared by reacting mixtures of PCL and PLA initiated by radical initiators [108]. The structure of the claimed PCL-*g*-PLA is unclear, there is probably a mixture of PCL-*g*-PLA, PLA-*co*-PCL, PCL-*g*-PCL, PLA-*g*-PLA and a mixture of PCL and PLA homopolymers. The final product is a compatibilizer of PCL and PLA blends, as evidenced by the DSC and DMA thermograms and SEM images. However, this compatibilizing effect is not significant, with only very small variations in melting temperature, melting enthalpy and variations in E’ and E’’ moduli observed.

1,4,8-trioxa [4.6]spiro-9-undecanone (TOSUO), a γ-acetal caprolactone, was polymerized—or copolymerized with CL—in the presence of Al(OiPr)_3_ in toluene at 25 °C to give poly TOSUO or P(TOSUO-*co*-CL) (Figure 20). CL and TOSUO have also been polymerized sequentially to give block copolymers [109]. The poly TOSUO is then deacetalyzed in the presence of triphenylcarbonium tetrafluoroborate to give a ketone, which is reduced in an alcohol by reaction with sodium borohydride to give a P(γ-hydroxy CL-*co*-CL) [110]. By reaction of the hydroxyl group with AlEt_3_, pendant OAlEt_2_ groups were obtained to initiate caprolactone, glycolide or lactide polymerizations and give PCL-*g*-PCL, PCL-*g*-PGA or PCL-*g*-PLA graft copolymers with grafts connected in γ-position of the CL unit.

Similarly, a γ-triethylsilyloxy-ε-caprolactone (TeSCL) was prepared by reaction of chlorotriethylsilane with 4-hydroxycyclohexanone followed by treatment with 3-chloroperbenzoic acid. TOSUO, CL and TeSCL were copolymerized to give a “hetero multifunctional” P(CL-*co*-TOSUO-*co*-TeSCL) (Figure 21) [47]. It should be noted that the lengths of poly TOSUO and poly TeSCL segments are rather short (DP ≅ 3). After a specific hydrolysis of the silyloxy groups to alcohols, the polymerization of CL initiated by the hydroxyl groups in the presence of Et_3_Al gives a poly[(CL-*co*-TOSUO)-*g*-CL)] copolymer. The acetal groups are then deprotected to alcohols which, after reaction with Et_3_Al, initiate the polymerization of D,L lactide to finally give a PCL grafted with both PCL and PLA segments.

Dai et al. recently prepared a PCL-*g*-(PCL-*b*-P(CL-Br)) bottlebrush using a thio-bromo “click” reaction on PCL-*b*-P(Br-CL) [111]. A P(CL-*co*-CL-N_3_) backbone was synthesized by ROP of CL and CL-Br monomers, followed by substitution of the pendant brominated groups with trimethylsilyl azide (TMS-N_3_). In parallel, the alkyne-terminated alkyne-PCL-*b*-P(CL-Br) was prepared by sequential ROP of CL and CL-Br using propargyl alcohol as the initiator. P(CL-*co*-CL-N_3_) and alkyne-PCL-*b*-P(CL-Br) were reacted by CuAAC click reaction to give PCL-*g*-(PCL-*b*-P(CL-Br). Finally, another thio-bromo “click” reaction of PCL-*g*-(PCL-*b*-P(CL-Br) on dimethylamino ethane thiol (DMAET) gives a cationic PCL-*g*-(PCL-*b*-P(CL-DMAET) copolymer (Figure 22).

Recently a PCL-*g*-PLA was prepared by the so-called “grafting through” approach, where a caprolactone substituted by a PLA chain was copolymerized with caprolactone (Figure 23) [112].

#### 3.3.3. PCL-*g*-Poly Lysine

Nottelet et al. prepared an amphiphilic polycation PCL-*g*-polylysine according to i) a “grafting from” method by polymerization of lysine N-carboxybenzyl (Z-lysine) initiated by the macropolycarbanion PCL^−^ and ii) a “grafting onto” of a preformed poly Z-lysine, previously activated by reaction with bromoacetyl chloride, on the macropolycarbanion PCL^−^ (Figure 24) [113]. Mechanisms for the polymerization of Z-lysine with PCL^−^ as macroinitiator are proposed. After the deprotection of the Z groups, amphiphilic copolymers were obtained that form micelle-like objects with various properties. In the grafting “onto” strategy two long polylysine chains (Mn = 60,000 g/mol) were grafted onto each PCL backbone giving large micelles (diameter = 500 nm). On the contrary, in the grafting “from” strategy the macropolycarbanion PCL^−^ can initiate many short polylysine chains, giving micelles of ≅100 nm diameter (Figure 25). The possibility to form polyplexes from these PCL-*g*-polylysine was evaluated through preliminary transfection tests to propose fully degradable and less toxic polycations [114].

#### 3.3.4. PCL-*g*-Poly N-Vinyl Caprolactam (PCL-*g*-PNVCLac) and PCL-*g*-Poly N-Vinyl Pyrrolidone (PCL-*g*-PNVP)

These copolymers were prepared from PCL-*co*-(αCL-CL) after substitution of the chlorine atom with a xanthate group. The xanthate enabled RAFT polymerization of N-vinyl caprolactam (NVCLac) or N-vinyl pyrrolidone (NVP) initiated with V-70 (2,2′-azobis(4-methoxy-2,4-dimethylvaleronitrile) (Figure 26) [115]. PNVCLac segments are water-soluble and heat sensitive. Similarly, a PCL-*g*-P(NVCLac-*co*-NVP) terpolymer was obtained by RAFT polymerization of a mixture of NVP and NVCLac. All these copolymers self-assemble in aqueous medium to give micelles whose diameter, LCST, thermosensitivity properties are function of the proportions of PNVCLac and PNVP segments.

A detailed micellization study was performed to investigate the thermosensitivity of the graft terpolymers with various compositions. It was shown that the presence of 10 or 20 mol% NVP units in the grafts of PCL-*g*-P(NVCL-*co*-NVP) copolymers leads to a shift of LCST from 32 °C to temperatures in the range of 38–40 °C.

#### 3.3.5. PCL-*g*-Dextran

Using PCL-yne obtained by the anionic modification method, Delorme et al. recently obtained PCL-*g*-dextran by reacting PCL-yne with a dextran activated at its chain end by an azide group (Figure 27) [112]. In aqueous medium, this amphiphilic copolymer forms micelles which were loaded with Doxorubicin (Dox), an anti-cancer drug. Dox-loaded PCL-*g*-Dextran micelles showed a specific efficacy toward colorectal cancer cells without affecting healthy cells. Moreover, the micelles were specifically internalized by colorectal cancer cells and not by healthy cells [34].

The main copolymers prepared from the PCL chain and literature references are summarized in Table 2.

## 4. Conclusions

PCL is one of the most studied biodegradable synthetic polymers for various applications in the biomedical field, especially for drug delivery and tissue engineering. This review focuses on the chemical modifications of the PCL backbone giving access to PCL-based graft copolymers. In all cases, the formation of PCL-based graft copolymers requires a functionalized PCL backbone that can be obtained by copolymerization of caprolactone with a pre-functionalized caprolactone or by direct functionalization of a preformed PCL via a chemical modification of its backbone. Once the functionalization is performed, the functional groups are used (i) to initiate polymerization of a comonomer by ATRP, ROP, RAFT, or ITP (grafting from method) or (ii) to react with an activated preformed polymer (grafting “on to” method). In addition to these approaches, the anionic modification of PCL provides a direct functionalization of the PCL backbone. The diverse graft copolymers produced by any of these methods are listed and the synthesis methods are described. The various types of graft copolymers are listed, and their principal characteristics and properties are described. Some physicochemical and biological properties of these new copolymeric systems are mentioned (bioavailability, drug loading and cell uptake, targeting and imaging properties, enzyme responsiveness, surfactant properties, and formation of stealth nanoparticles) These properties mainly concern the amphiphilicity, the formation of micelles with low CMC, the interactions with active agents or the compatibilization of mixtures, leading mainly to applications in the biomedical field.

## Figures and Tables

**Figure 1 molecules-27-07339-f001:**
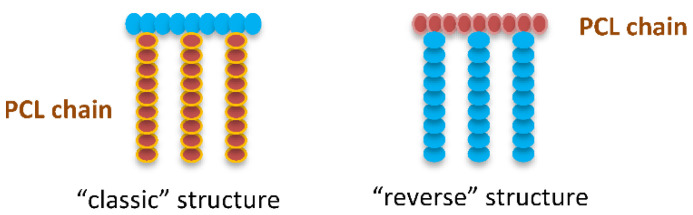
Schematic illustration of “classic” and “reverse” structures.

**Figure 2 molecules-27-07339-f002:**
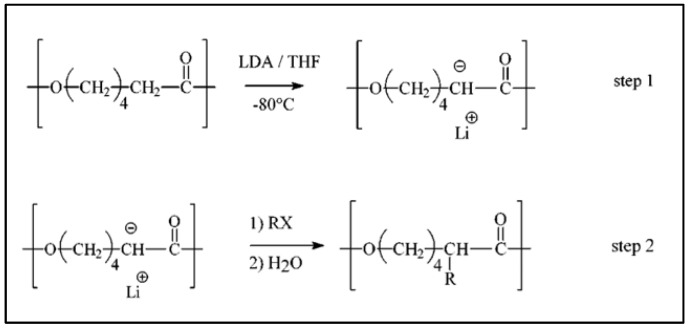
Scheme of the post-polymerization method via anionic reaction (from Ponsart et al. [55]). Copyright American Chemical Society, reproduced with permission.

**Figure 3 molecules-27-07339-f003:**
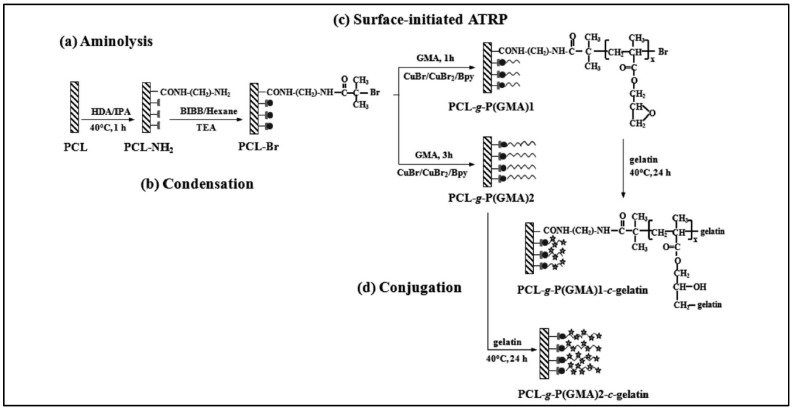
Surface-initiated ATRP method to obtain PC-*g*-P(GMA)1-c-gelatin and PCL-*g*-P(GMA)2-c-gelatin. The various steps are shown: (**a**) aminolysis of the surface (**b**) condensation (**c**) ATRP (**d**) conjugation (from Xiong et al. [65]). Copyright Royal Society of Chemistry, reproduced with permission.

**Figure 4 molecules-27-07339-f004:**
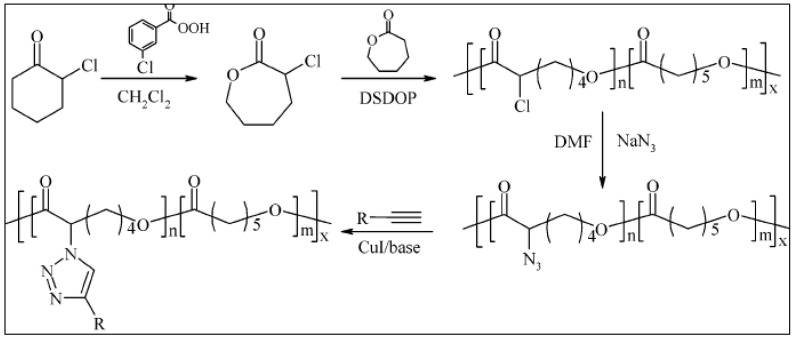
Synthesis scheme of PCL-*g*-PEG by click chemistry (R = MeO-PEG) (from Riva et al. [43]). Copyright American Chemical Society, reproduced with permission.

**Figure 5 molecules-27-07339-f005:**
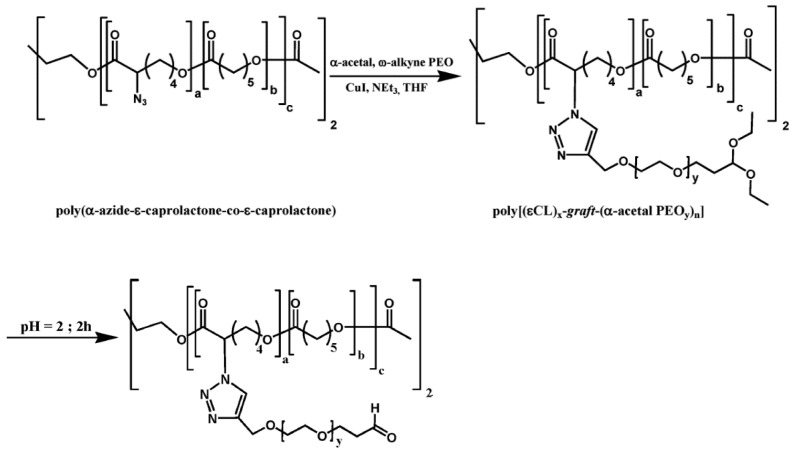
Synthesis scheme of PCL-*g*-[(α-Mannose PEG)] (from Freichel et al. [81]]). Copyright American Chemical Society, reproduced with permission.

**Figure 6 molecules-27-07339-f006:**
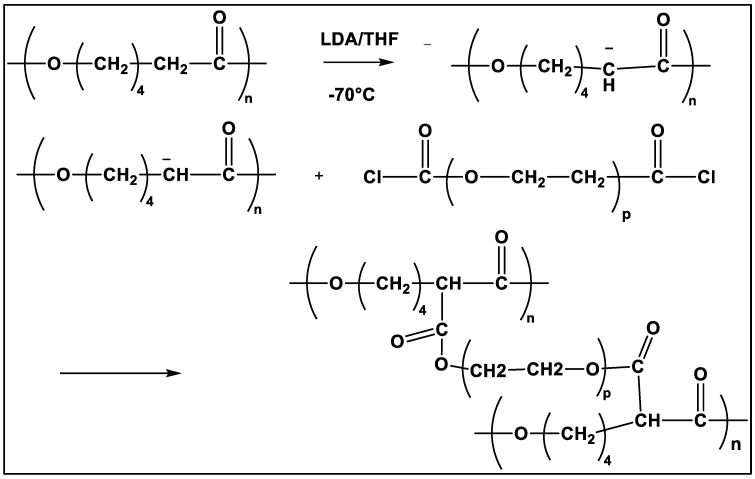
Grafting reaction of PEG dichloride on the macropolycarbanion PCL^−^ (only modified units are presented) (from Coudane et al. [83]). Copyright Wiley-VCH GmbH. R, reproduced with permission.

**Figure 7 molecules-27-07339-f007:**
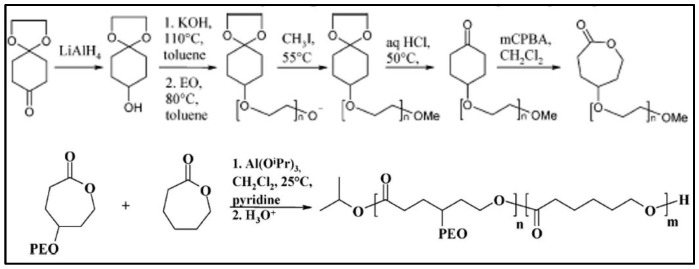
Synthesis scheme of P(ε-CL-*co*-γ-PEG-CL)) (from Rieger et al. [87]).) Copyright American Chemical Society, reproduced with permission.

**Figure 8 molecules-27-07339-f008:**
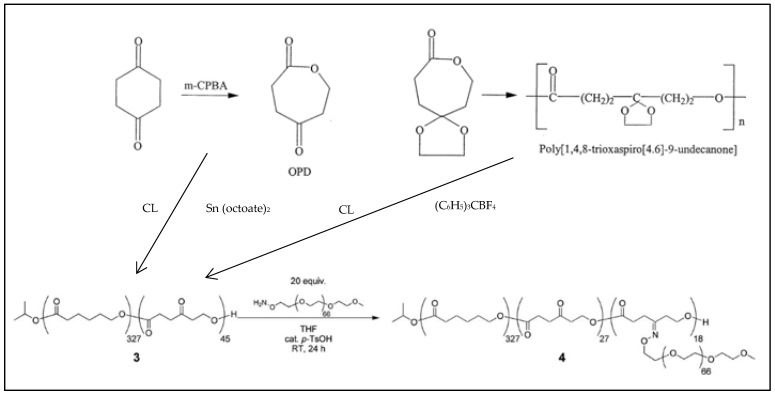
Synthesis scheme of PCL-*g*-PEG with a substitution in γ-position of the CL unit (adapted from Latere et al. [90] and Iha et al. [91]]).

**Figure 9 molecules-27-07339-f009:**
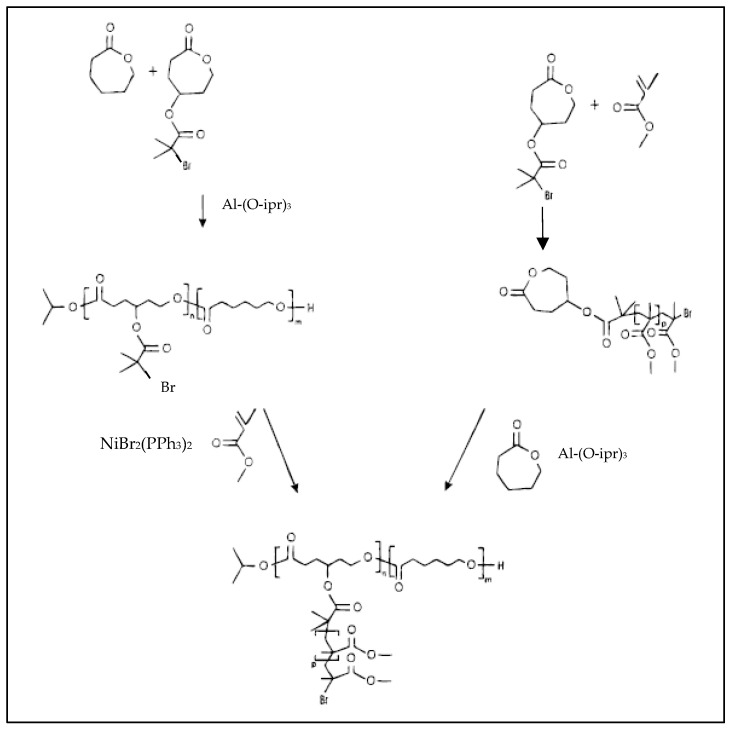
Two synthetic approaches of PCL-*g*-PMMA (from Mecerreyes et al. [46]). Copyright American Chemical Society, reproduced with permission.

**Figure 10 molecules-27-07339-f010:**
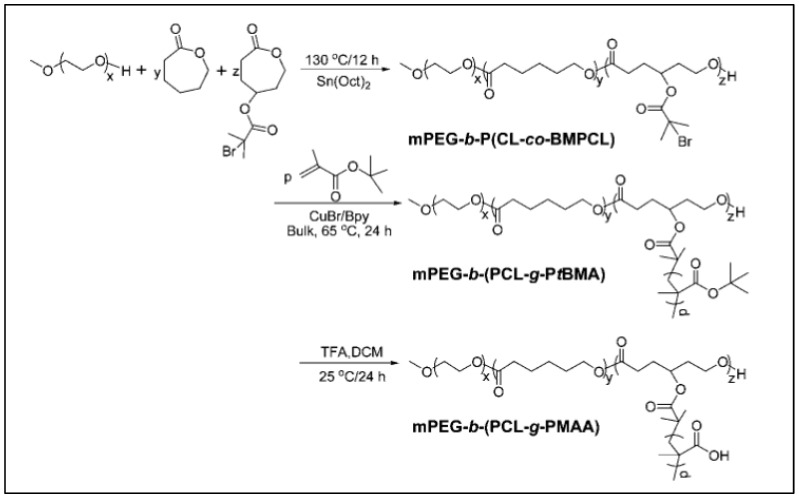
Synthesis routes of MeOPEG-*b*-(PCL-*g*-PMAA) (from Chang et al. [92]). Copyright Royal Society of Chemistry, reproduced with permission.

**Figure 11 molecules-27-07339-f011:**
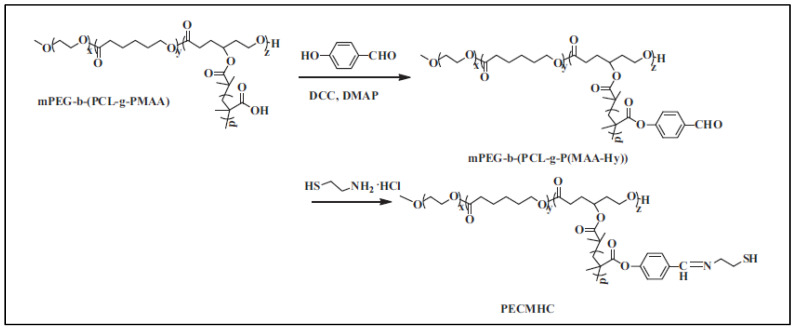
Synthesis route of PECMHC (from Deng et al. [93]). Copyright Elsevier, reproduced with permission.

**Figure 12 molecules-27-07339-f012:**
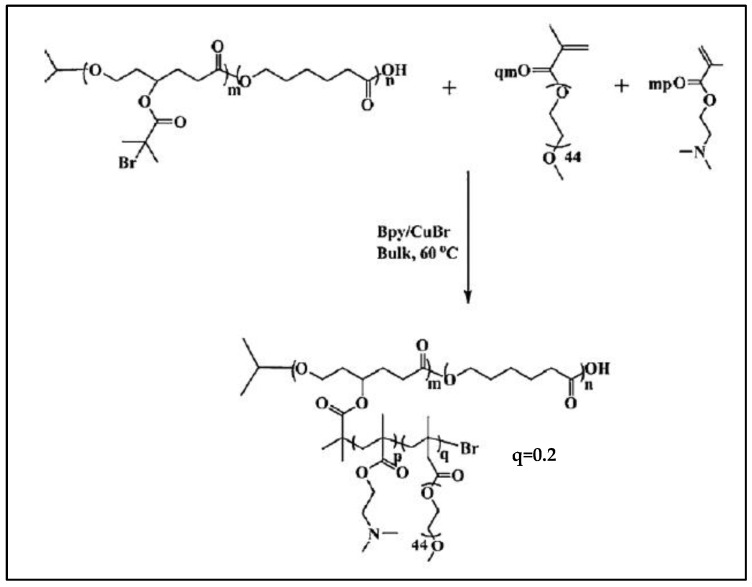
Synthesis scheme of PCL-*g*-P(DMAEMA-*co*-MeOPEGMMA). (from Guo et al. [98]). Copyright Royal Society of Chemistry, reproduced with permission.

**Figure 13 molecules-27-07339-f013:**
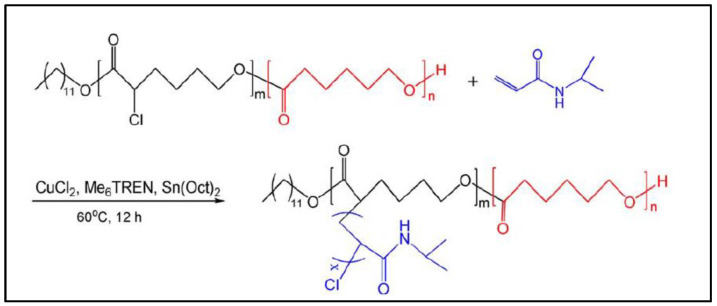
Synthesis scheme of PCL-*g*-PNIPAAm (from Li et al. [100]). Copyright Wiley-VCH GmbH. R, reproduced with permission.

**Figure 14 molecules-27-07339-f014:**
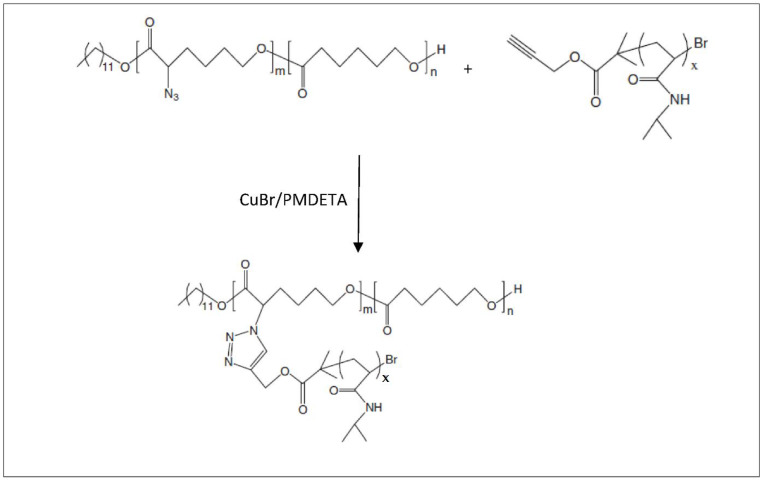
Synthesis scheme of MeOPCL-*g*-PNIPAAm (from Li et al. [102]). Copyright Wiley-VCH GmbH. R, reproduced with permission.

**Figure 15 molecules-27-07339-f015:**
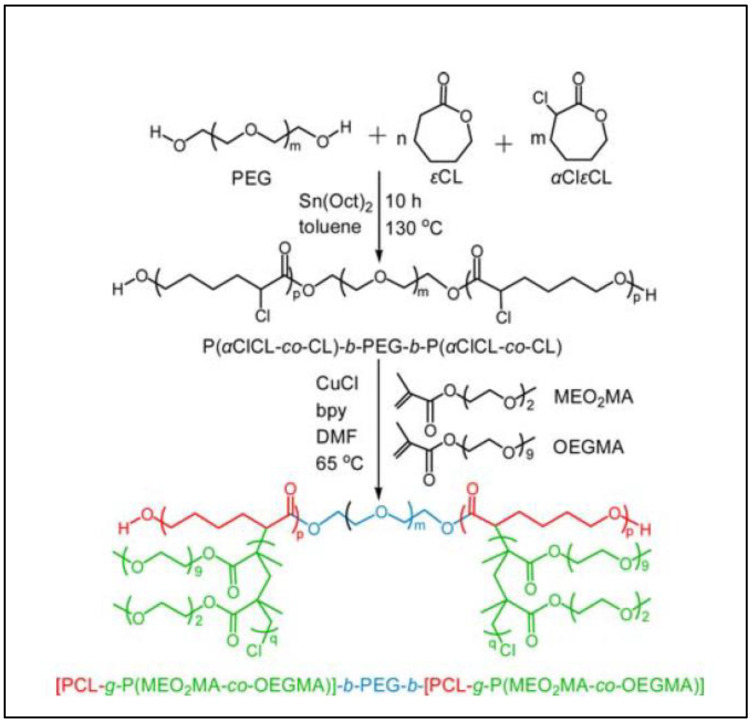
Synthesis of PCL-*g*-P(MEO2MA-*co*-OEGMA)]-b-PEG-b-[PCL-*g*-P(MEO_2_MA-*co*-OEGMA) (from Wang et al. [103]). Copyright Springer, reproduced with permission.

**Figure 16 molecules-27-07339-f016:**
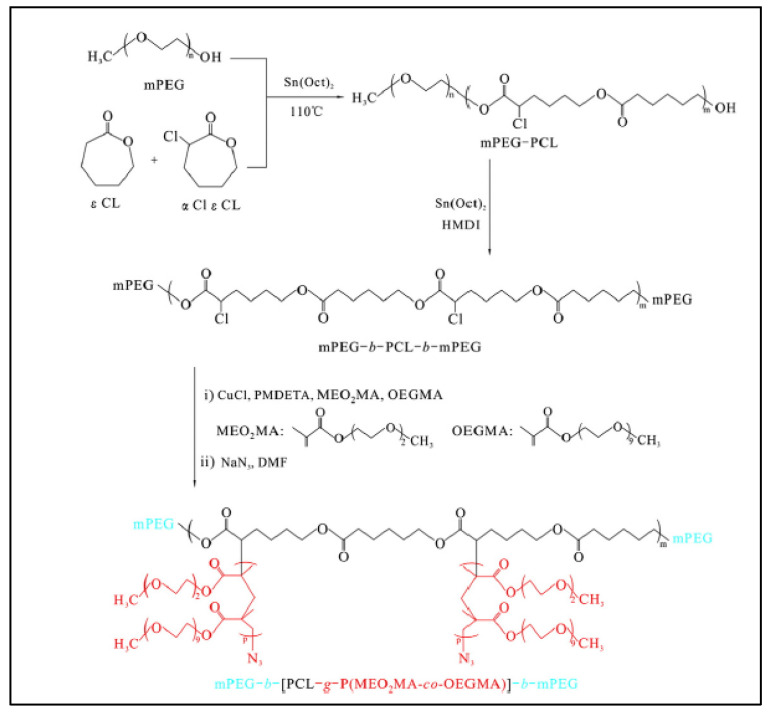
Synthesis scheme of PEG-b-[PCL-*g*-P(MEO2MA-*co*-OEGMA)]-b-PEG (tBG2) (according to An et al. [104]). Copyright Elsevier, reproduced with permission.

**Figure 17 molecules-27-07339-f017:**
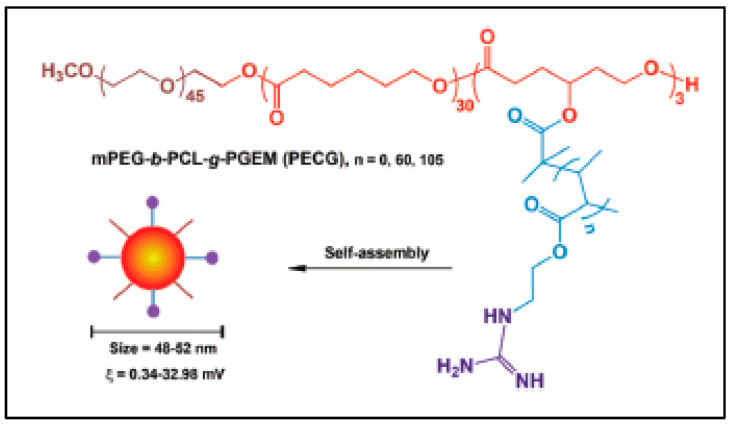
Chemical structure of MeOPEG-*b*-PCL-*g*-PGEM (PECG) copolymers and the schematic diagram of copolymer self-assembly (from Li et al. [105]). Copyright Royal Society of Chemistry, reproduced with permission.

**Figure 18 molecules-27-07339-f018:**
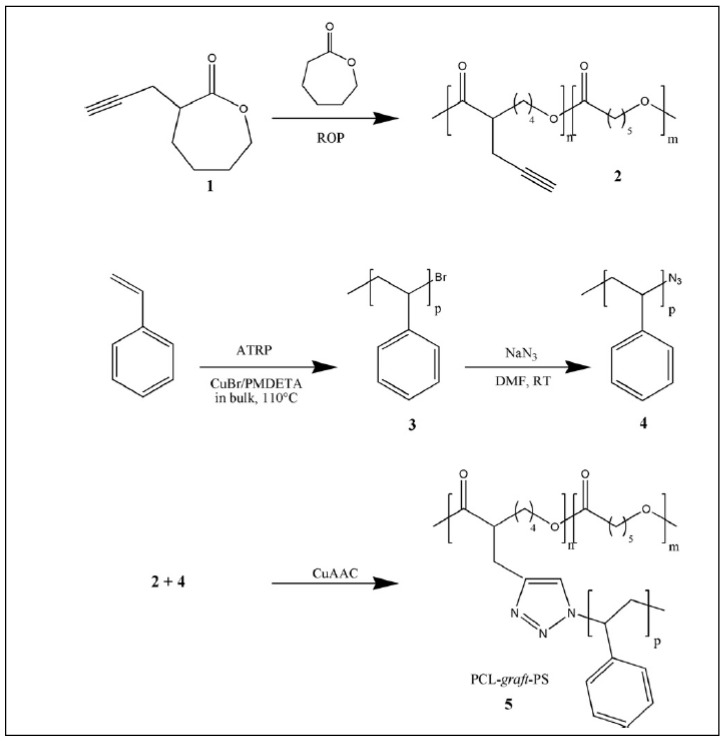
Synthesis of PCL-*g*-PS (from Darcos et al. [106]). Copyright Elsevier, reproduced with permission.

**Figure 19 molecules-27-07339-f019:**
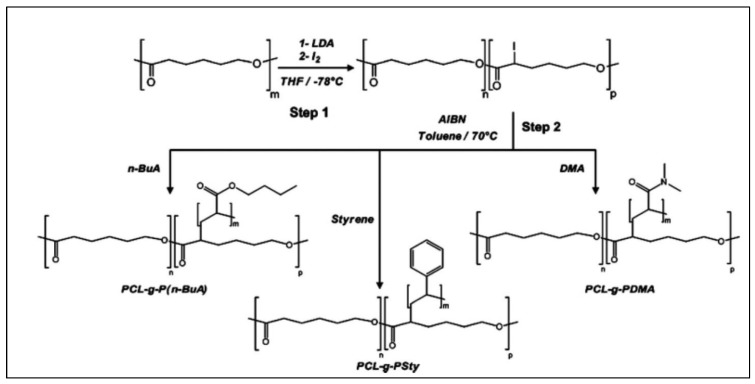
Syntheses of PCL-*g*-PS, PCL-*g*-P(n-BuA) and PCL-*g*-PDMA by ITP initiated by iodized-PCL (from Nottelet et al. [104]). Copyright Wiley-VCH GmbH. R, reproduced with permission.

**Figure 20 molecules-27-07339-f020:**
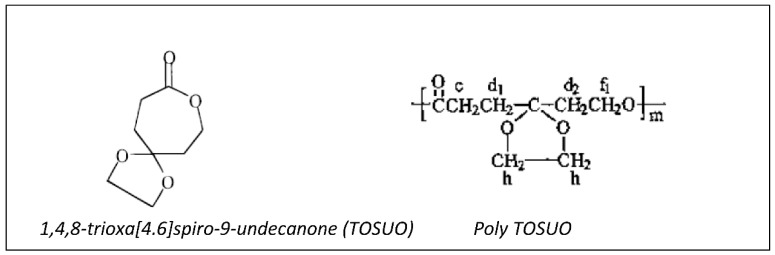
Structures of TOSUO and poly TOSUO.

**Figure 21 molecules-27-07339-f021:**
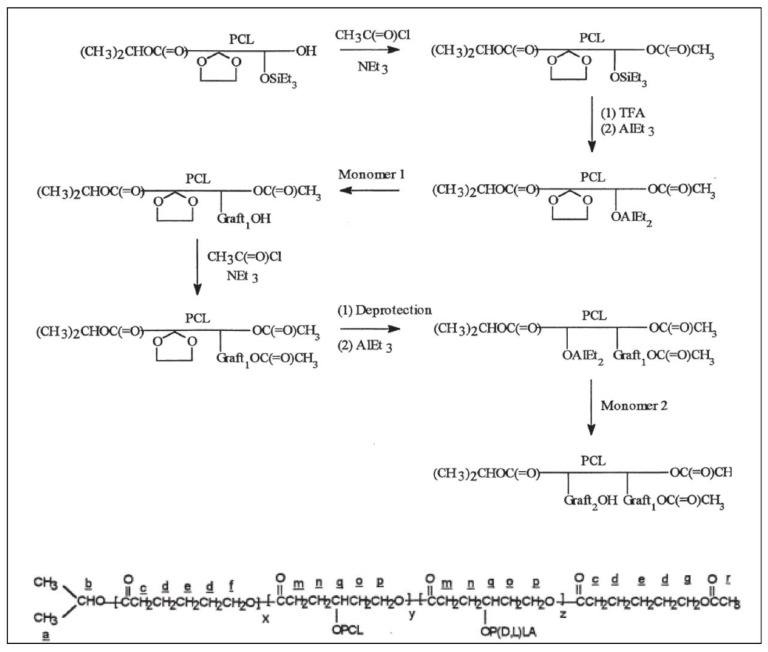
Synthesis of PCL grafted with both PCL and PLA segments (from Stassin et al. [44]). Copyright Wiley-VCH GmbH. R, reproduced with permission.

**Figure 22 molecules-27-07339-f022:**
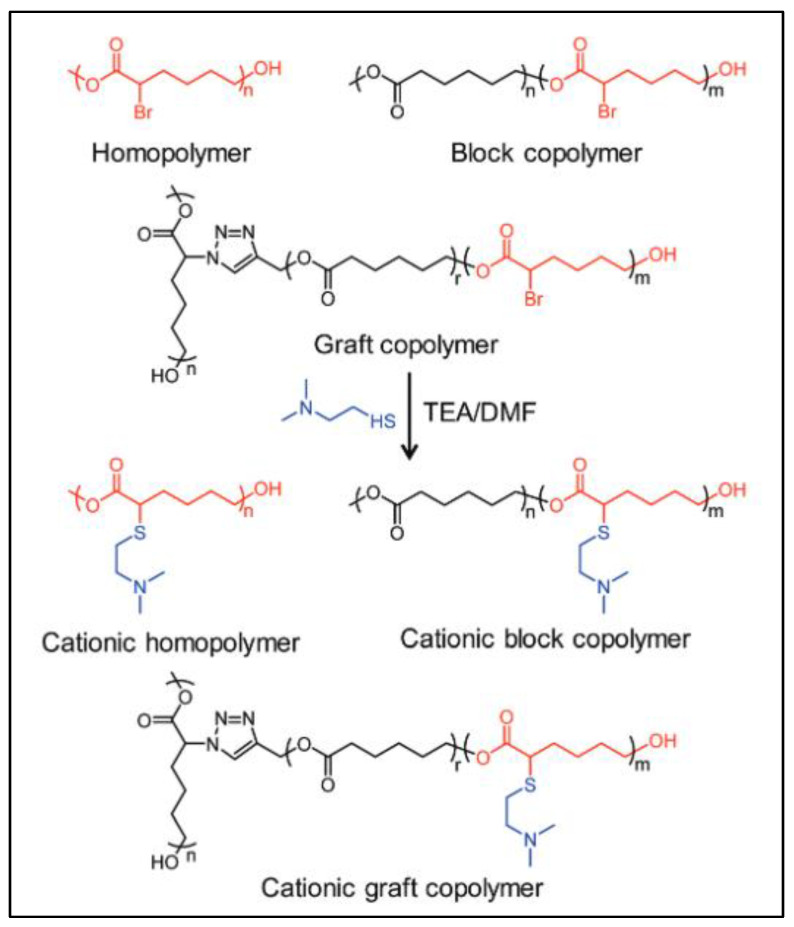
Structure of cationic copolymers based on PCL-*g*-(PCL-b-P(CL-DMAET) (from Dai et al. [108]). Copyright Royal Society of Chemistry, reproduced with permission.

**Figure 23 molecules-27-07339-f023:**
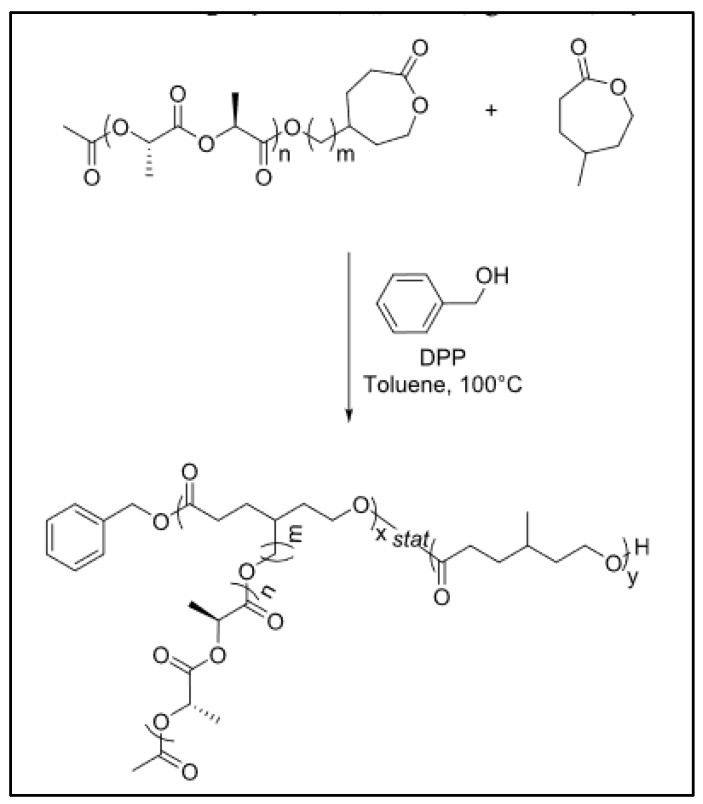
Synthesis of a PCL-*g*-PLA according to a “grafting through” method (from Fournier et al. [112]).

**Figure 24 molecules-27-07339-f024:**
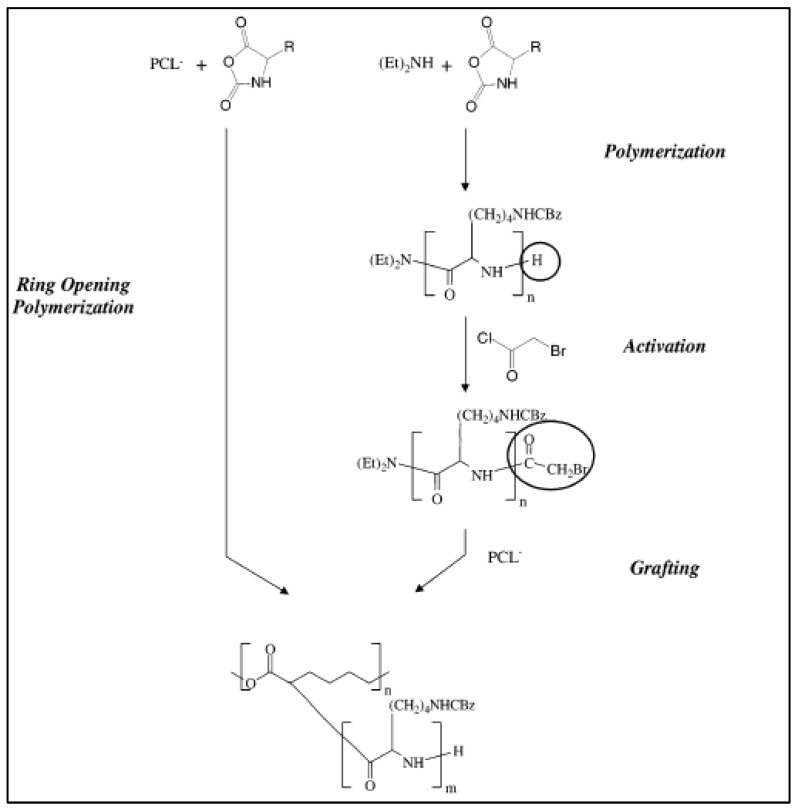
Synthesis of PCL-*g*-Poly Z-lysine via the grafting” on to” and ”from” strategies (from Nottelet et al. [109]). Copyright American Chemical Society, reproduced with permission.

**Figure 25 molecules-27-07339-f025:**
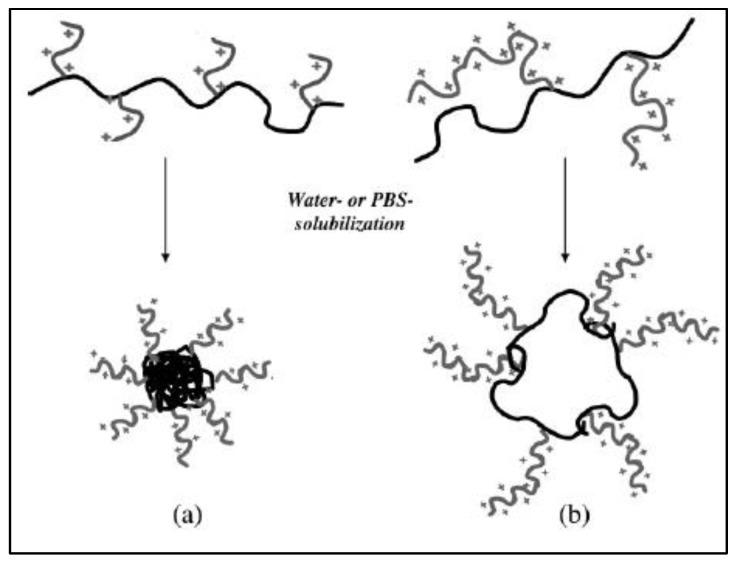
Schematic representation of micelles of PCL-*g*-PolyLysine prepared by (**a**) grafting “from” or (**b**) “onto” methods (from Nottelet et al. [109]). Copyright American Chemical Society, reproduced with permission.

**Figure 26 molecules-27-07339-f026:**
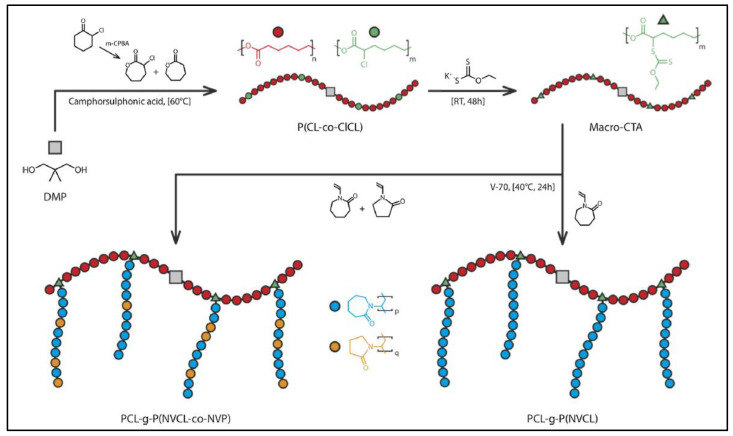
Schematic representation of the synthesis route of PCL-*g*-PNVCL and PCL-*g*-(PNVCL-*co*-NVP) (from Winninger et al. [111]). Copyright Elsevier, reproduced with permission.

**Figure 27 molecules-27-07339-f027:**
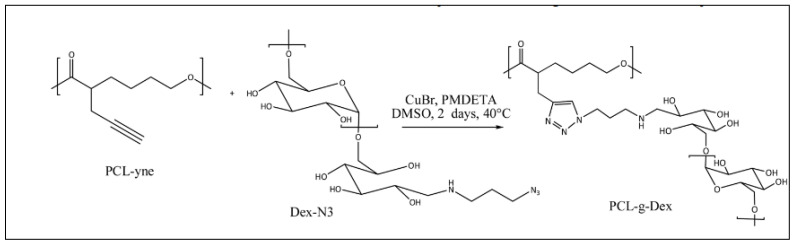
Reaction scheme of the synthesis of PCL-*g*-dextran (from Delorme et al. [112]). Copyright Elsevier, reproduced with permission.

**Table 1 molecules-27-07339-t001:** CMC and hydrodynamic diameters of various PCL-*g*-PEG (adapted from Zhang et al. [80]) Copyright Wiley-VCH GmbH, reproduced with permission.

Substitution Ratio of PEG (%)	CMC (mg/L)	Hydrodynamic Diameter (nm)
5	1.82	41.5
10	3.09	18.8
30	30.2	15.6
50	83.2	10.8

**Table 2 molecules-27-07339-t002:** Main PCL-based graft copolymers according to the literature.

Type of Graft Copolymer	Starting Compound	Grafting of Polymeric Segments	Ref.
PCL-*g*-PEG			
	P(α-ClCL)	-NaN_3_ -CuAAC of Alkyne-PEG	[79,80,81]
	PCL-yne	CuAAC of MeOPEG-N_3_	[42]
	PCL-yne	Anionic modification	[60,84,85]
	P(MeOPEG-γCL)		[88,90]
	P(2-oxepane-1,5-dione)	Reaction of ω-amino MeOPEG	[91]
PCL-*g*-poly (meth)acrylate derivatives			
	P(α-ClCL)	ATRP of MMA	[79]
		ATRP of N-isopropylacrylamide	[101,102]
	BMPCL	ATRP of MMA	[46]
		ATRP of tBuMA	[92]
		ATRP of MMA	[92,93]
		ATRP of n-BuA	[92,93]
		ATRP of DMAEMA	[95]
		ATRP of DMAEMA and MeOPEGMMA	[95,96,97]
		Azide substitution of MeOP(αCl-CL-*co*-CL)	[102]
		ATRP of MEO_2_MA and OEGMA	[103,104,105]
		-ATRP of (*t*-butoxycarbonyl) amino ethyl methacrylate -Guanidilation	[105]
	PCL-I	ITP of n-butyl acrylate	[107]
		ITP of N,N-dimethyl acrylamide	[107]
PCl-*g*-PS			
	P(α-ClCL)	-Azide substitution of MeOP(αCl-CL-*co*-CL) -reaction of propargyl bromoisobutyrate -ATRP of styrene	[43]
	PCL-yne	Azido-PS	[106]
	PCL-I	ITP of Styrene	[107]
PCL-*g*-PCL PCL-*g*-PLA	Poly TOSUO	-P(γ-hydroxy CL-*co*-CL) -ROP polymerization of CL or lactide	[110] [47]
	BMPCL	-CuAAC of P(CL-*co*-CL-N_3_) and alkyne-PCL-*b*-P(CL-Br) -ATRP of styrene	[111]
PCL-*g*-Poly lysine	PCL^−^	Polymerization of Z-lysine	[113]
		Reaction of activated poly lysine	[113]
PCL-*g*-PNVCLac	P(α-ClCL)	-Substitution by xanthate -RAFT polymerization of NVCLac	[114]
PCL-*g*-PNVP	P(α-ClCL)	-Substitution by xanthate -RAFT polymerization of NVP	[115]
PCL-*g*-dextran	PCL-yne	CuAAC of Dextran-N_3_	[34]

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
