# Peer review of "Poly(ε-caprolactone)-Based Graft Copolymers: Synthesis Methods and Applications in the Biomedical Field: A Review"

_molecules, 2022, doi:10.3390/molecules27217339_

Round 1

Reviewer 1 Report

The authors are requested to make the manuscript clear toward the goal of the review topic. Section 2 is haphazard and needs more attention.

Reviewer 2 Report

This paper mainly covers the synthesis and applications of PCL based copolymers and their applications within the biomedical field. The authors list various copolymers and discuss each of them in detail. Overall, this is a good review paper regarding this specific topic.

However, here are some problems/questions/comments regarding this draft:

1.       The authors covered both grafting from and grafting onto methods for copolymer synthesis. However, there is another commonly used synthesis method of "grafting through", which is not discussed within this paper. Not sure if there are previous studies covering that method. Please comment on it and add the relevant discussions within the draft.

2.       The authors claimed the benefits of PCL based graft copolymers are useful and important between line 65-73. However, the only reference cited there ([19]) is a review paper of PLA based graft copolymer written by the authors. I do not think there are enough background studies to support that argument. Since this is one of the main merits the authors tried to claim, please cite some more relevant studies and carry out the discussion in more detail. Current discussion is not sufficient.

3.       Between line 73-74, the authors claimed three pathways are followed. However, I do not really see a clear explanation of the three pathways. Please improve that.

4.       Between line 86-88, these two methods correspond to the grafting from and grafting onto methods mentioned within the abstract. However, it seems the authors did not mention that again within the draft. Please use those specific terms to better describe the methods discussed within the draft.

5.       Following on from the last point, since two different approaches can be applied to the PCL based graft copolymers, it would be better if the authors separate the synthesis accordingly in section 3. Current discussion mixed both methods in one section, which is not the best way to do that.

6.       Format for “Figure 6” in line 340 is not consistent.

7.        At line 411, it should be “CMC” instead of “CAC”

8.       In table 2, it seems the fonts are not consistent. Please double check that.

9.       At line 677, I would assume it should be “CMC” instead of “CAC”

10.   For some of the schemes for the reactions shown within the draft, the qualities are relatively low (such as Figure 1, Figure 13, Figure 20 ). I would recommend the authors replot these figures to improve their qualities.

11.   There are multi “Error! Reference source not found” within the draft. Please double check and avoid that.

12.   For the grafted copolymer, actually they are a kind of bottlebrush polymers. A short discussion of the overall properties should be covered within the introduction section and references should be included as: https://doi.org/10.1039/C4CS00329B ; https://doi.org/10.1002/chem.201900520;

https://doi.org/10.1021/acs.macromol.9b01801; https://doi.org/10.1021/acs.macromol.8b02366; https://doi.org/10.1021/acs.biomac.8b01171;  

https://doi.org/10.1021/acsmacrolett.0c00384; https://doi.org/10.1039/C8SM01127C

13.   Some formattings of the citations seem inconsistent. For example, at line 99, it should be “[32,34]” instead of “[32], [34]”. Please double check the formats.

Reviewer 3 Report

The authors have prepared a review article and discussed the synthesis, characterizations, and biomedical application of PCL-based graft copolymers. Upon surface functionalization, PCL shows homogeneous solubility in aqueous solutions and organic solvents, low toxicity, and high pharmaceutical efficiency. The current mini-review draft demonstrated recent studies of such surface-functionalized PCL for biomedical applications.

The reviewer recommends this work be published after a minor review.

The reviewer has the following comments

1.      The quality of all the figures is poor and the authors must add high-resolution images

2.      Although, the author mentioned biomedical applications. However, they did not discuss them in detail. They must cover some biomedical applications of PCL such as drug delivery, imaging, sensing, and therapeutic drugs.

3.      A separate section of future prospective should be added to the revised manuscript

4.      The following sentence is repeated more than 25 times in the manuscript, it should be removed.

‘’Error! Reference source not found’’

5.      The introduction is missing some very essential references. For instance, the following references are missing, which are covered a broad range of synthetic polymers including PCL. Thus, the following articles should be quoted in the introduction and other relevant sections

https://doi.org/10.1021/acsbiomaterials.2c00786 https://doi.org/10.1016/j.compositesb.2022.110150  

It would be more realistic to cover such kind of research work in the current manuscript. Which will enrich the quality of the current manuscript as well as the inquisitiveness of the readers.

6.      According to the new information, the conclusions should be updated in the revised manuscript
